# Morphology and Wear Resistance of Composite Coatings Formed on a TA2 Substrate Using Hot-Dip Aluminising and Micro-Arc Oxidation Technologies

**DOI:** 10.3390/ma12050799

**Published:** 2019-03-08

**Authors:** Shaopeng Wang, Lian Zhou, Changjiu Li, Zhengxian Li, Hongzhan Li

**Affiliations:** 1School of Materials Science and Engineering, Xian Jiaotong University, Xi’an 710049, China; zhoul@c-nin.com (L.Z.); licj@mail.xjtu.edu.cn (C.L.); 2Corrosion and Protection Center, Northwest Institute for Nonferrous Metal Research, Xi’an 710016, China; lzxqy725@163.com (Z.L.); tylhz0042@163.com (H.L.)

**Keywords:** hot-dip aluminising, micro-arc oxidation, titanium

## Abstract

Aluminium layers were coated onto the surface of pure titanium using hot-dip aluminising technology, and then the aluminium layers were in situ oxidised to form oxide ceramic coatings, using the micro-arc oxidation (MAO) technique. The microstructure and composition distribution of the hot-dip aluminium coatings and ceramic layers were studied by using scanning electron microscopy and energy-dispersive X-ray spectroscopy. The phase structure of the MAO layers was studied using X-ray diffraction. The surface composition of the MAO layer was studied by X-ray photoelectron spectroscopy. The wear resistance of the pure titanium substrate and the ceramic layers coated on its surface were evaluated by using the ball-on-disc wear method. Therefore, aluminising coatings, which consist of a diffusion layer and a pure aluminium layer, could be formed on pure titanium substrates using the hot-dip aluminising method. The MAO method enabled the in-situ oxidation of hot-dip pure aluminium layers, which subsequently led to the formation of ceramic layers. Moreover, the wear resistance values of the ceramic layers were significantly higher than that of the pure titanium substrate.

## 1. Introduction

Titanium and its alloys have attracted increasing attention in the aviation, aerospace, chemical, biomedical and other sectors, due to their high specific strength, excellent corrosion resistance and good biocompatibility [1,2]. However, titanium exhibits low hardness and poor wear resistance, and when it is used as a friction member, its surface wear can cause component failure [2,3,4,5,6,7]. Since coating can improve the surface wear resistance of titanium alloys and improve the service life of titanium alloy parts, the preparation of wear-resistant coatings on the surface of titanium and titanium alloys has become a hot research topic. Various surface methods are used to prepared wear-resistant coatings on the surfaces of titanium and titanium alloys. For example, thermal spray technology, which is used to prepare Al_2_O_3_, Al_2_O_3_-TiO_2_ and Al_2_O_3_-TiB_2_ ceramic coatings on the surface of titanium alloys, can significantly improve their wear resistance [3,8,9]. Physical vapour deposition (PVD) of diamond-like carbon (DLC), TiN and TiSiCN nanocomposite coatings have also been used to improve the wear resistance of titanium alloys, and remarkable results have been achieved [4,6,10,11]. Combinations of composite-processing techniques could also be used to improve the wear resistance of titanium and its alloys, to overcome the limitations of single technologies. For example, an Al_2_O_3_-TiO_2_ coating was deposited onto the surface of a titanium substrate using the thermal spraying technique, and then, electron beam remelting was used to improve the density and wear resistance of this coating [5]. A TiO_2_ and DLC composite coating was deposited on the surface of a titanium alloy using micro-arc oxidation (MAO) and magnetron sputtering. The surface roughness and porosity of the micro-arc oxidised layer were reduced by the DLC coating, therefore improving the surface hardness. Moreover, the wear resistance of the coating deposited using MAO and magnetron sputtering was superior to that of the coating deposited using MAO alone [12].

Thermal spray ceramic coating is an effective method that is used to improve the wear resistance of titanium alloys. However, the efficiency of thermal spraying coating is low, the spraying equipment is expensive, and the process of preparing uniform coatings on complex workpiece surfaces is complicated. Furthermore, thermal spraying cannot be used to deposit coatings onto the surfaces of small inner bores. Therefore, the adaptability of the thermal spraying technology is limited. While the PVD of nitrides, DLC, carbide ceramics and other coatings is also an effective method to improve the surface wear resistance of titanium alloys, it should be performed in a vacuum chamber, the deposition equipment is complex, and the size of the workpieces that could be subjected to this treatment is limited by the size of the vacuum chamber. 

The MAO method can be used in situ to deposit oxide ceramic coatings onto the surfaces of valve metals (Ti, Al, Zr, Mg, Nb and Ta) [13,14]. This method has recently become a research hotspot. Compared with other process technologies, this method is simple and adaptable, and it does not require complicated equipment, and as long as the power supply is adequate, an uniform ceramic coating can be deposited on surfaces of large workpieces. During the MAO process, a ceramic coating could be formed by the in situ oxidation of a matrix material component, or a mixed ceramic coating may be generated by the co-oxidation of the matrix material and the electrolyte component. For example, during the MAO of the surface of aluminium alloy substrates, the aluminate system is used to form a ceramic layer, which generally consists of Al_2_O_3_, while silicates generate mixed ceramic coatings composed of Al_2_O_3_ and silica [15,16,17]. The MAO process generally produces ceramic coatings on the surface of titanium alloys. These coatings mainly consist of TiO_2_, and they can be incorporated into ceramics such as zirconia by regulating the electrolyte [18,19,20]. In situ MAO ceramic coatings are also one of the methods used to improve the wear resistance of titanium alloys. However, while TiO_2_-based ceramic coatings can be generated by using single MAO processes, because the hardness of TiO_2_ is relatively inferior to that of Al_2_O_3_, the wear resistance of titanium and titanium alloys would be smaller than that of Al_2_O_3_. In addition, since the arcing voltage of titanium substrates during the MAO process is high, the voltage required for the oxidation of titanium substrates is higher than that which is required for the aluminium substrates. Therefore, the coatings formed on the surface of titanium substrates would be less dense than those that obtained during the MAO of aluminium, and the wear resistance values of the titanium-based coatings would be inferior to those of the Al_2_O_3_-based coatings. If Al_2_O_3_-based coatings could be deposited on the surface of titanium alloys using MAO, the wear resistance of titanium alloys could be improved, owing to the high hardness and the relatively good compactness of Al_2_O_3_, which are obviously superior to those of TiO_2_-based coatings.

During hot-dip aluminising, workpieces are immersed into molten liquid aluminium and a series of physical and chemical reactions occur on their surfaces. After the workpieces were extracted from the molten aluminium and cooled, aluminium coatings are formed on their surfaces. This method is advantageous owing to its simple process, low cost, and high maturity, and has been used for the large-scale production of hot-dip aluminised coatings on steel parts for more than 100 years. Additionally, the hot-dip technology is mainly used to form Fe–Al intermetallic compounds and pure aluminium layers on the surface of steel parts, and to improve the high temperature oxidation resistance, corrosion resistance, and wear resistance of steel [21,22,23,24,25,26,27]. After steel undergoes hot-dip aluminising, the hot-dip aluminium layer on its surface is oxidised to form an Al_2_O_3_ ceramic coating, using the MAO method. This further improves the high-temperature performance and wear resistance of steel, and it has achieved good research. Therefore, the wear resistance and high-temperature oxidation resistance of steel that was subjected to MAO after hot-dip aluminising have been reported to be significantly higher than those of steel that only underwent hot-dip coating [28,29,30,31]. Although hot-dip aluminising of titanium has been studied, researchers have mainly focused on using hot-dip aluminising to form Ti-Al intermetallic compound phases on the surface of titanium, to improve the oxidation resistance of titanium alloys [32,33,34,35,36,37,38,39,40]. However, a few scholars have studied the formation of Al_2_O_3_ ceramic coatings on the surface of titanium that has undergone hot-dip aluminising using MAO. Therefore, the goal of this study was to use hot-dip aluminising and MAO to indirectly generate Al_2_O_3_ ceramic coatings on the surface of titanium and titanium alloys, thus, improving their wear resistance.

## 2. Materials and Methods

The substrate material, TA2, was industrial pure titanium, and its nominal composition was (mass percentages): Fe ≤ 0.3, C ≤ 0.1, N ≤ 0.05, H ≤ 0.015, O ≤ 0.25 and Ti being the balance. The material was processed into a sample that was 60 mm × 12 mm × 1.5 mm in size. The hot-dip coating material was industrial pure aluminium, and its nominal composition was (mass percentages): Si ≤ 0.08, Fe ≤ 0.1, Cu ≤ 0.01 and Al balance.

The samples were first cleaned using a metal cleaning agent, to remove the surface oil, and then pickled to remove the surface oxide scale, quickly rinsed with alcohol, dried and immediately subjected to hot-dip aluminising. The MAO experiment was performed shortly after the hot-dip aluminising was completed.

The hot-dip aluminising experiment was carried out in a self-made well-type resistance furnace Φ300 mm × 500 mm in size. The temperature was controlled by using an automatic temperature meter with an accuracy of ±1 °C, and the rated temperature was 1100 °C. For the hot-dip aluminising process, industrial pure aluminium was mechanically cut into small pieces, cleaned using a metal cleaning agent, degreased using absolute ethanol, dried, added to an alumina crucible and placed in a well resistance furnace. The specific process parameters of hot-dip aluminising are listed in Table 1. During the hot-dip aluminising process, the surface of the molten liquid aluminium was covered using a mixture of 46% KCl, 28% NaCl and 26% Na_3_AlF_6_ (mass percentages) to prevent the oxidation of the surface of the liquid aluminium. After hot-dip aluminising, the samples were lifted from the liquid aluminium at a rate of 5 cm/s and solidified in air.

Immediately after hot-dip aluminising, the samples were placed in a bath containing the electrolyte for the MAO treatment. A direct current pulse power supply was used for the MAO process. The electrolyte was an aqueous solution of sodium silicate (Na_2_SiO_3_), sodium hexametaphosphate (Na_6_O_18_P_6_) and a small amount of sodium hydroxide (NaOH), which was mainly used to adjust the pH of the solution. The parameters of the MAO process are listed in Table 1. During the MAO process, the temperature of the electrolyte was maintained under 40 °C, using a cooling system. After the MAO experiment was completed, the sample was rinsed using deionised water, dried and stored in a desiccator.

A tungsten filament scanning electron microscopy (SEM, TESCAN VEGAII XMU, Brno, Czech Republic) device was used to observe the morphology of the coatings. An Oxford X-sight energy X-ray dispersive spectrometry (EDS, Oxford INCA, Oxford, UK) detector was used to analyse the composition of the coatings. The phase structure of the ceramics layers was analysed using a X-ray diffraction (XRD, D/max-2200pc, RIGAKU, Tokyo, Japan) instrument. The XRD analysis conditions were as follows: the anode consisted of a copper target, the diffraction angle, 2θ, ranged from 20° to 90°, the scanning rate was 2°/min, the electron acceleration voltage was 40 kV and the current was 40 mA. An X-ray photoelectron spectroscopy (XPS, ESCALAB 250Xi, Thermo Fisher Scientific, Waltham, MA, USA) apparatus was utilised to analyse the composition of the MAO coatings. The wear resistance and friction coefficient of the MAO coatings were tested using an MS-T3000 friction and wear tester (MS-T3000, Lanzhou Institute of Chemical Physics, Lanzhou, China). The test conditions were as follows: the wear-testing ball was a silicon nitride ball 5 mm in diameter, the loading pressure was 0.98 N, the friction diameter was 6 mm and the wear test time was 20 min. After the wear tests were completed, the cross-section geometry of the wear scars was analysed using the step meter feature of the material surface comprehensive performance tester (MFT-4000, Lanzhou Institute of Chemical Physics, Lanzhou, China), the wear volumes were calculated using the geometric profile, and the wear resistance values were estimated using the wear volumes.

The wear volumes of the samples were calculated by using the following formula:(1)$$V_{w}=\frac{2\pi \cdot t\cdot r}{6b}\left(3t^{2}+4b^{2}\right),$$
where *Vw* is the wear volume (mm^3^), *t* is the wear scar depth (mm), *b* is the wear scar width (mm), and *r* is the wear track radius (mm).

## 3. Results

### 3.1. Microscopic Characteristics of Hot-Dip Aluminised Layer

Figure 1 illustrates the surface morphology secondary electrons (SE) images and the EDS spectra of the hot-dip aluminised layers formed on the surfaces of the titanium substrate and their elemental compositions obtained using EDS analysis. It can be seen from Figure 1 that the hot-dip aluminised layers that were formed on the surfaces of the titanium substrates were continuous and uniform and that the surface morphologies were wrinkled, due to grain boundary shrinkage. The surface topography of the T-2 sample was more complex than that of the T-1 sample. A small number of micropores were observed in Figure 1a, while a relatively large number of micropores and some microcracks were observed in Figure 1c. The EDS spectra and composition analysis of the surface coating components indicated that the surfaces of the hot-dip aluminised layers consisted entirely of elemental aluminium.

Figure 2 represents the grain boundary diagram and grain size distribution obtained by image analysis, using the surface morphology images of Figure 1a,c. The grain boundaries are marked by green lines using the image analysis method, and the size and distribution of the crystal grains are analysed by image analysis. It can be seen intuitively from Figure 2 that the grain size of the T-2 sample (Figure 1b) was significantly larger than that of the T-1 sample (Figure 1a). Due to the irregular shape of the grains, in order to visually compare the sizes of the crystal grains, the sizes of the crystal grains were characterized by area. Figure 2c,d are bar graphs of the grain area distribution obtained by image analysis. The number of crystal grains obtained in Figure 2c was 57, and the number of crystal grains obtained in Figure 2d was 44. It can be seen that the grain area of the T-1 sample was mostly below 1000 μm^2^, there are 48, and the area of a small number of grains exceeds 1000 μm^2^, there are nine, and the largest grain area was 1762 μm^2^ and corresponding to the grain size was 65 μm in length and 42 μm in width. The grain area below 1000 μm^2^ of the T-2 sample is significantly reduced, there are 31, and there are 13 above 1000 μm^2^, and the maximum grain area was over 3000 μm^2^, its area was 3430 μm^2^, and the corresponding grain size is 123 μm in length, and 42 μm in width.

Figure 3 illustrates the cross-section back-scattered electrons (BSE) microscopic images of the hot-dip aluminising layers. The hot-dip aluminium layers exhibited a two-layered structure, where the outer layer was a pure aluminium layer, and the inner layer was a diffusion layer. It can be seen that the diffusion layer was well-bonded to both the pure aluminium layer and the titanium substrate, and no microscopic defects, such as cracks or holes, could be observed at the interface between the diffusion layer and the substrate, or that between the diffusion and pure aluminium layers. The aluminium layer obtained after 3 min of hot-dip aluminising at 710 °C (sample T-1, Figure 3a,b, where Figure 3b is a partial enlargement of Figure 3a) was dense and continuous, and it presented no defects, such as cracks or micropores. The thickness of the overall hot-dip aluminium coating (including the pure aluminium and diffusion layers) was measured by using SEM (Figure 3a). The average thickness of the hot-dip aluminium coating was calculated to be approximately 17.9 μm, by averaging the three measured thickness values: the highest, lowest and intermediate values. Similarly, the thickness of the diffusion layer was measured using the range observable in Figure 2b, and the average thickness of the diffusion layer was calculated to be approximately 0.6 μm. The density of the pure aluminium layer obtained after 3 min of hot-dip aluminising at 750 °C (sample T-2, Figure 3c,d, where Figure 3d is a partial enlargement of Figure 3c) was relatively smaller than that of the hot-dip aluminium layer obtained at 710 °C. The hot-dip aluminium layer that was formed on the surface of the T-2 sample possessed a small number of defects, such as micropores. Furthermore, the overall average thickness of this hot-dip aluminium coating was approximately 19.9 μm, and the average thickness of the diffusion layer was approximately 1.5 μm. The overall thickness of the hot-dip aluminium layer of the T-2 sample was slightly higher than that of the T-1 sample. Additionally, the thickness of the diffusion layer of the T-2 sample was significantly higher than that of the T-1 sample.

Figure 4a,b depict the EDS element mapping analysis results of aluminium and titanium on the surfaces of the T-1 and T-2 samples after hot-dip aluminising. Figure 4c,d represent the EDS point-scan analysis, and an average value at three points of the diffusion layer of the hot-dip aluminised samples, as shown in Figure 3b,d. It can be seen that the presence of titanium was not detected in the pure aluminium layers of the T-1 and T-2 samples. The titanium content of the diffusion layer gradually increased, while the aluminium content gradually decreased from the interface between the diffusion and the pure aluminium layers to the interface between the diffusion layer and the substrate. Owing to the high thickness of the diffusion layer of the T-2 sample, the content of aluminium decreased, and the content of titanium increased relatively slowly. It can be seen from Figure 4c,d that the average atomic percentages of aluminium and titanium in the diffusion layer of the T-1 sample were 76.19 and 23.81%, respectively, while the corresponding percentages in the diffusion layer of the T-2 sample were 65.76 and 34.24%, respectively. This indicated that the temperature of the hot-dip aluminising process significantly affected the diffusion layer composition.

### 3.2. Microscopic Properties of the MAO Layer

Figure 5 presents the surface morphology SE images and the surface composition analysis results of the samples after the hot-dip aluminising coating formed on the TA2 substrate was subjected to MAO treatment. Both hot-dip aluminised samples underwent the same MAO process. It can be seen that the surface morphologies of the ceramic layers were typical of MAO ceramic layers, and the morphologies of the craters formed by the MAO discharge could be clearly observed. As can be seen from Figure 5a,b, after MAO, the surface discharge holes of the T-1 sample were smaller than those of the T-2 sample, and the surfaces of both samples were relatively dense. Since the same MAO process was used for both hot-dip aluminised samples, the difference in the surface compactness was mainly attributed to the differences in the parameters of the hot-dip aluminising processes. It can be seen from the EDS surface scan results in Figure 5b,d that the MAO layer contains five elements: Al, Si, O, Na and P, and the oxygen contents of the two samples were very close. While the aluminium content of the ceramic layer of the T-1 sample subjected to MAO was slightly higher than that of the T-2 sample that underwent MAO, the silicon content was slightly lower. The atomic percentage contents of the aluminium and silicon of the MAO layer of the T-1 sample were 28.11 and 15.97%, respectively, and the corresponding aluminium and silicon percentages of the MAO layer of the T-2 sample were 23.57% and 19.80%, respectively. Since both samples were subjected to the same MAO process, the main reason for the differences in the aluminium and silicon contents of the surface layers was attributed to the differences in the hot-dip aluminium layers of the samples. The amounts of sodium and phosphorus detected in the MAO layers were very small, and they mainly originated from the residual electrolyte in the coating.

Figure 6 illustrates the image analysis results of the size and distribution of the surface discharge holes of the MAO layer, using the surface topography photograph of Figure 5a,c. Figure 6a,b were distributions of the actual positions and sizes of the holes obtained by image analysis. In the figures, red is the actual position, size and distribution of the holes. Figure 6c,d were the bar graphs of the area distribution of the discharge holes. Since the discharge holes are not circular in size, the area of the holes obtained by image analysis was used to characterise of the size of the holes. As is apparent from Figure 6a,b, the discharge holes were randomly distributed throughout the MAO layer. The discharge holes of the MAO layer of the T-2 sample were more than the T-1 sample, and they were larger in size. In the Figure 6c (the T-1 sample), the number of holes obtained by image analysis was 335, the area of most of the holes was below 4 μm^2^, and the area of only a small number of holes exceeded 4 μm^2^. The area of the largest hole was 8.7 μm^2^, and the corresponding hole had a size of 3.6 μm in length and 3.0 μm in width. In Figure 6d (T-2 sample), the number of holes was 536, the number of holes below 5 μm^2^ was 411, and the number of holes exceeding 5 μm^2^ was 125. The area of the largest hole was 38.0 μm^2^, and the size of the corresponding hole was 11.6 μm in length and 7.7 μm in width. It can be clearly seen from the size distribution of the holes that the number of discharge holes and the sizes of the T-2 sample were significantly larger than those of the T-1 sample.

Figure 7 illustrates the cross-section BSE microscopic images of the hot-dip aluminised samples after MAO treatment. It can be seen that the obtained coatings exhibited a three-layered structure: the top layer was the ceramic layer, the second layer was the unoxidised pure aluminium layer, and the innermost layer was the diffusion layer that was obtained by hot-dip aluminising. A part of the pure aluminium layer was oxidized to form a ceramic layer, while the rest of the pure aluminium layer and the diffusion layer were kept unoxidised. The surfaces of the ceramic layers were rough, and they exhibited a certain degree of undulation. The average thickness was calculated by using the measured maximum, minimum and median thickness values. As shown in Figure 7a, the average thickness of the ceramic layer and the unoxidised pure aluminium layer with an interdiffusion zone was respectively calculated to be 16.9 μm and 14.1 μm. In Figure 7c, the average thickness of the ceramic layer was calculated to be 21.0 μm, using the same method, and that of the unoxidised pure aluminium, together with the interdiffusion layer was 13.9 μm. Compared with Figure 3a,c, the MAO layer in Figure 7a,c was approximately four times thicker than that of the consumed pure aluminium. Moreover, it can be seen from Figure 7b,d that the MAO layer was well-bonded to the pure aluminium layer, and no obvious microscopic defects, such as holes and cracks, were observed at the interface. The interface between the ceramic and pure aluminium layers was clear, but it was undulating. This implied that the growth rate of the ceramic layer was different at the ceramic/aluminium interface. Additionally, a number of holes were observed in the cross-sections of the ceramic layers, and this was determined by the nature of the MAO process. However, the difference between the substrate material and the MAO process will affect the distribution and size of these holes.

Figure 8 represents the cross-sectional EDS line-scan analysis of the combined ceramic and pure aluminium layers of the composite coating. As can be seen, the contents of sodium and phosphorus were low, and sodium and phosphorus were not detected when scanning the section line. It can be seen that silicon was present in the ceramic layers of the two samples, and its distribution in the area between the surface of the sample and the interface of the ceramic and pure aluminium layers was substantially gentle. The aluminium content of the area between the surface of the sample and the interface of the ceramic and pure aluminium layers was significantly lower than that of the pure aluminium layer, and it was basically the same as that of the ceramic layer. Since no diffusion layer existed between the ceramic and pure aluminium layers, no significant changes were detected in the contents of the other elements.

Figure 9 illustrates the surface XRD patterns of the oxide layers formed by MAO. The ceramic layers formed by MAO mainly consisted of α-Al_2_O_3_, γ-Al_2_O_3_, mullite and unoxidised aluminium. The contents of Al_2_O_3_ and mullite of the T-2 sample were slightly higher than those of the T-1 sample. The diffraction peak of aluminium could be observed in the XRD pattern of the ceramic layer, and that could be attributed to the aluminium in the ceramic layer not being completely oxidised during MAO, or the unreacted pure aluminium layer.

To further determine the surface phase composition of the MAO layers, XPS analysis was performed on the MAO layer on the surface of the T-1 sample, and the results are presented in Figure 10. The Si2p peak occurred at the binding energy of 103.45 eV, which indicated that silicon mainly existed as SiO_2_ in the MAO layer. The Al2p peak occurred at the binding energy of 74.82 eV, which indicated that aluminium in the coating mainly existed as Al_2_O_3_. This further demonstrated that the oxide layer mainly consisted of mullite and Al_2_O_3_ phases.

### 3.3. Friction and Wear Properties

To evaluate the wear resistance of the composite coatings formed on the surface of the pure titanium substrate using hot-dip aluminising followed by MAO, the friction and wear characteristics of the TA2, T-1 and T-2 samples were evaluated by using the ball-on-disc wear method. Figure 11 depicts the friction curves of the three samples obtained from the friction and wear test. It can be seen that the friction coefficient gradually increased in time for all three samples. After approximately 15 min, the friction coefficient became almost constant, and while small fluctuations were still observed, the overall trend was rather stable. During the stable phase, the friction coefficient of the TA2 sample was approximately 0.52, while those of the T-1 and T-2 samples: 0.62 and 0.65, respectively, were higher. Therefore, forming ceramic layers on the surface of pure titanium by combining the hot-dip aluminising and MAO techniques, led to an increase in the friction coefficients of the samples.

Figure 12 represents the cross-sectional geometrical outline of the wear scars obtained after the friction and wear test measured using a step meter. It can be clearly seen that the wear scar of the TA2 sample was significantly deeper and wider than those of the ceramic layer samples; therefore, indicating that the ceramic layers significantly improved the wear resistance of pure titanium. Table 2 lists the measured depth and width values of the wear scars and the wear volumes, calculated according to formula (1). Based on the data in Table 2, it could be concluded that the wear resistance values of the ceramic layers formed using MAO were significantly better than that of the pure titanium sample. The wear scar of the T-1 sample was slightly deeper, and its width was slightly smaller, compared with those of the T-2 sample, while the wear volumes of the two specimens were very close. Figure 13 allows to compare the wear volumes of the T-1, T-2 and TA2 samples.

## 4. Discussion

Hot-dip aluminising involves three processes, the first is the aluminium liquid wetting substrate, followed by the interdiffusion of the substrate elements and the aluminium element to form a solid intermetallic compound, and finally, the sample is lifted from a molten aluminium bath, and the liquid state aluminium rapidly cooled and solidifies to form a pure aluminium layer [26,33,35,37]. In the process of hot-dip aluminising on the TA2 substrate, the aluminium liquid must first wet the TA2 substrate. According to Lin et al. [35], liquid aluminium exhibited good wettability for titanium at temperatures above 600 °C, and the wetting properties could be attributed to the reactive wetting system. In general, the surfaces of metal substrates are covered by thin oxide films, and the pure titanium matrix is no exception. Owing to its very high affinity for oxygen, when the surface-activated pure titanium substrate is exposed to the atmosphere, a thin oxide film forms on its surface, which affects the wettability of liquid aluminium for pure titanium. However, during hot-dip aluminising, pure titanium samples, which are initially maintained at room temperature are rapidly immersed into high-temperature liquid aluminium. Owing to the heat transfer from the liquid aluminium, the surface temperature of the pure titanium samples increases rapidly. Since the thermal expansion coefficient of the oxide film and pure titanium do not match, the generated thermal stress causes the cracking of the thin oxide film on the surface of pure titanium. When liquid aluminium penetrates the rupture site, it comes in contact with the pure titanium matrix and wets it (Figure 14a) [41,42]. The infiltrated aluminium atoms react with titanium atoms to form Ti–Al intermetallic compounds that diffuse into the titanium substrate [21,35]. The growth of the intermetallic compounds causes further cracking and shedding of the oxide film on the surface of the titanium substrate, and the wettability is further improved. The complete wetting of the titanium substrate by the aluminium liquid is the key to preparing the hot-dip aluminized layer. Only when the aluminium liquid completely infiltrates the titanium substrate, will the interdiffusion of the aluminium element and the titanium element form a diffusion layer. After the diffusion layer was formed, when the sample is lifted from the aluminium liquid, the aluminium liquid adheres to the surface of the substrate and rapidly solidifies to form a pure aluminium layer. A complete aluminium layer was observed from the surface topography of Figure 1 and the diffusion layer and pure aluminium layer with good interfacial adhesion were observed in Figure 3, indicating that the aluminium liquid has properly wetted the TA2 substrate under the process conditions used herein.

According to the phase diagram of the Ti–Al binary alloys, as the concentration of aluminium increases, four intermetallic compounds can be detected in the 710–750 °C temperature range: Ti_3_Al, TiAl, TiAl_2_ and TiAl_3_. During the initial stage of hot-dip aluminising, the TiAl_3_ intermetallic compound, which featured the lowest concentration of titanium, was first formed, and a small number of aluminium atoms were solid-dissolved into the titanium substrate to form a solid solution with titanium [37], as illustrated in Figure 14b. As the hot-dip time increased, the aluminium atoms continued to diffuse into the titanium substrate owing to the increase in temperature and concentration gradient, while titanium diffused into liquid aluminium. The TiAl3 intermetallic compound, which formed on the surface, further reacted with titanium to form TiAl2. Since TiAl2 was an unstable compound, it further reacted with titanium atoms to form TiAl, which presented a lower aluminium concentration that TiAl2. Moreover, TiAl2 also reacted with the aluminium atoms that diffused into the diffusion layer to form TiAl3. Furthermore, the generated TiAl compound reacted with the titanium atoms that entered the diffusion layer and formed Ti3Al, which featured a lower aluminium concentration than TiAl, and then TiAl3 reacted with titanium, therefore repeating the transformation process: TiAl3 → TiAl2 → TiAl → Ti3Al. During the hot-dip aluminising process, as the immersion time increased, the solid/liquid interface continued to migrate into liquid aluminium, and the thickness of the diffusion layer gradually increased. The aluminium-rich TiAl3 phase was located near the solid/liquid interface of the diffusion layer; the aluminium-depleted Ti3Al phase and the solid titanium–aluminium solution were found in the vicinity of the diffusion layer/ titanium substrate interface of the diffusion layer, while TiAl3, TiAl2, TiAl, Ti3Al and other titanium-aluminium intermetallic compounds were located at the centre of the diffusion layer [43,44,45,46,47], as shown in Figure 14c. Titanium and aluminium elements form a gradient distribution, as shown in Figure 4. The growth of the diffusion layer is affected by the hot-dip aluminising time and temperature, and temperature is the main influencing factor. As the temperature increases, the elementals will gain greater diffusion power, especially in the case of titanium elements with larger atomic numbers, which are more likely to diffuse into the solid–liquid interface, resulting in the acceleration of diffusion layer growth [21,23,27,35,36]. It can be observed from Figure 4 that under the same hot-dip aluminising time, the temperature increases, the thickness of the diffusion layer increases significantly, and the content of titanium element in the diffusion layer also increases significantly.

At the end of the hot-dip aluminising process, when the sample was lifted from liquid aluminium, the liquid aluminium that adhered to the surface of the solid sample, owing to its viscosity, was rapidly solidified, and it formed a pure aluminium layer. During the formation of the pure aluminium layer, the molten aluminium liquid contacts the room temperature air to rapidly cool and solidify. The aluminium crystal grains rapidly nucleate and grow, and the grain boundary finally solidifies. Due to the cooling shrinkage of the aluminium liquid, the surface of the pure aluminium layer is observed to have obvious shrinkage at the grain boundary, and the grain boundary is concaved. Also, due to the existence of shrinkage stress, there are defects such as holes and cracks at the grain boundary [48,49,50]. As shown in Figure 1 and Figure 2, the higher the hot-dip temperature, the more pronounced the shrinkage phenomenon, which is why there are more defects in the 750 °C hot-dip aluminizing layer (T-2 sample). In addition, the hot-dip temperature is high and the crystal grains are more likely to grow. Therefore, the crystal grain of the aluminium layer at 750 °C during hot-dip aluminisation is relatively large. The thickness of the pure aluminium layer was determined by the lifting rate of the sample from the liquid aluminium and the temperature of the liquid aluminium; the lifting rate was the main factor. In this study, the same lifting rate was used for both processes; therefore, the thicknesses of the pure aluminium layers of the samples were very similar.

After hot dip aluminizing was completed, a two-layer hot-dip aluminizing layer structure, composed of a pure aluminium layer and an intermetallic compound diffusion layer having a gradient distribution between the titanium and aluminium elements, was formed on of the TA2 substrate, as shown in Figure 3.

During MAO, the applied electric field between the anode workpiece and the cathode plate produces a high-voltage discharge. The high-voltage discharge breaks down the initial oxide film on the surface of the anode to generate a plasma micro-arc discharge, which generates high temperature, melts the substrate alloy in the discharge micro-domain, and vaporises the surrounding liquid to generate extremely high pressure. High temperatures and high pressures cause the original oxide film on the surface of the substrate to undergo crystalline structure transformation. At the same time, the oxygen ions and other ions from the electrolyte also enter the micro-arc zone through the discharge channel, and a plasma chemical reaction occurs with the molten substrate. The reaction product is deposited on the inner wall of the discharge channel, where it forms an oxide layer. As the action of the applied electric field continues, the above process always takes place in the weak areas of the oxide film, and micro-discharges continue to occur. Finally, an oxide ceramic coating is formed in situ on the surface of the anode sample [13,15,16,17,19,20].

For the hot-dip aluminised layers formed on the surface of the pure titanium substrate, the oxide ceramic coating was only generated on the surface of the pure aluminium layer during the MAO process, while the role of the titanium matrix was only to conduct current. There was no essential difference in principle from the overall pure aluminium sample. The formation of its oxide ceramic layer was not fundamentally different from the overall aluminium sample. However, since the hot-dip aluminising process is different than the MAO one, the compactness of the pure aluminium layers would be different. Defects, such as microscopic holes (Figure 1 and Figure 3), in the pure aluminium layer obtained by hot-dip aluminising at 750 °C resulted in an increase of the current density during the actual MAO process. Macroscopically, the two hot-dip aluminised samples, T-1 and T-2, were subjected to the same MAO process. However, during the actual MAO process, owing to the presence of microscopic defects in the T-2 sample, the breakdown voltage and current of the discharge micro-region were actually higher than those of the T-1 sample, and the discharge hole in the formed oxide layer of the T-2 sample was larger than that of the T-1 sample. This was the main reason for the differences in the microscopic morphologies of the MAO layers of the two hot-dip aluminised samples (Figure 5 and Figure 6) [15,19,31]. In addition, owing to the high discharge voltage of the micro-region, the oxidation process was carried out more fully, which also explained that the content of the ceramic phase of the MAO layer of the T-2 sample was higher than that of the T-1 sample. In this study, a silicate electrolyte system was used for MAO. According to the growth mechanism of the MAO coating, the elemental silicon in the electrolyte participated in the oxidation reaction; therefore, silicon oxide was formed in the resulting coating [15,16,19,29], as shown in Figure 9 and Figure 10. 

After the micro-arc oxidation, a composite coating of a three-layer structure composed of a diffusion layer, an unoxidised pure aluminium layer and a ceramic layer formed by micro-arc oxidation was finally formed on the TA2 substrate.

As shown in Table 2, Figure 12 and Figure 13, under the same friction and wear test conditions, the wear volume of the two prepared ceramic layer samples was relatively close, which is about one-tenth that of the TA2 sample, indicating that the wear resistance was significantly improved. The main reason is that the hard ceramic phase, such as alumina and mullite improved the wear resistance. The wear scar width of the T-1 sample was slightly smaller, and the depth was slightly deeper. This is mainly related to the microscopic characteristics of the ceramic layer. As with before the analysis, the actual micro-discharge voltage and current of the T-1 sample during the micro-arc oxidation process were lower. Under the same oxidation time, the content of the hard ceramic phase formed was lower than that of the T-2 sample. The unoxidised pure aluminium content was relatively large. In the case of friction and wear, the ground silicon nitride ball was more easily pressed, so the depth of the wear scar was large. However, because the discharge hole is small, the coating was relatively dense, and the coating was not easily brittle, due to the friction of the grinding ball, so the wear scar width was slightly smaller. The actual micro-area discharge voltage and current of the T-2 sample are high, and the discharge holes formed are large and numerous, and the coating compactness was poor. During the friction and wear process, the repeated contact friction of the grinding ball easily leads to the detachment of the brittle ceramic phase. Therefore, the wear scar width was slightly wider. However, due to its high ceramic phase content, the friction ball was not easily pressed, so its wear marks were shallow.

A ceramic coating containing aluminium and silicon oxides can be prepared on the surface of a pure titanium substrate, by a combination of hot-dip aluminising and micro-arc oxidation processes, using a silicate system electrolyte. The micro-arc oxidation process directly results in micro-arc oxidation on the titanium, and its alloys can only prepare ceramic coatings of titanium oxide and silicon oxide, and the coating growth efficiency and coating compactness are lower than that of the aluminium micro-arc oxidation. In addition, because the hardness of titanium oxide is lower than that of aluminium oxide, the coating wear resistance is not as good as that of the coating, which is mainly composed of aluminium oxide [51,52].

The pure aluminium layer is first deposited onto the surface of titanium by magnetron sputtering, and then the deposited aluminium layer is oxidized to form a ceramic coating by micro-arc oxidation technology. It can also prepare composition coating that is composed of a pure aluminium layer and a ceramic layer on the titanium surface. However, compared with the process adopted herein, the deposition coating has no diffusion layer, the coating does not form a metallurgical bond, and the magnetron sputtering deposition needs to be completed in a vacuum chamber, and the preparation process is relatively complicated. The deposition efficiency of the coating is also lower [53].

## 5. Conclusions

1) An aluminium coating was deposited onto the titanium substrate, using the hot-dip aluminising technique, and it presented a two-layered structure with a pure aluminium layer outside, and a diffusion Ti–Al layer inside. The Ti–Al diffusion layer was well-bonded to the pure aluminium layer and the pure titanium substrate;

2) A ceramic layer mainly consisting of α-Al_2_O_3_, γ-Al_2_O_3_, mullite and unoxidised aluminium was obtained after the treatment of the hot-dip aluminised layer. The thicknesses of the aluminium layers consumed during MAO were approximately one quarter of the thicknesses of the obtained ceramic layers. Moreover, the content of the ceramic phase in the ceramic layer obtained after the 750 °C hot-dip aluminised sample underwent MAO was higher than that of the 710 °C hot-dip aluminised sample. The interface between the ceramic and residual aluminium layers was well combined;

3) The wear resistances of the samples were significantly improved after ceramic layers were formed on the surfaces of the pure titanium substrates. The friction coefficient of the pure titanium sample was determined to be 0.52, while the friction coefficients of the samples subjected to MAO after hot-dip aluminising at 710 and 750 °C were 0.62 and 0.65, respectively. The corresponding calculated wear volumes of the three samples were, respectively, calculated to be 0.736, 0.076 and 0.074 mm^3^.

## Figures and Tables

**Figure 1 materials-12-00799-f001:**
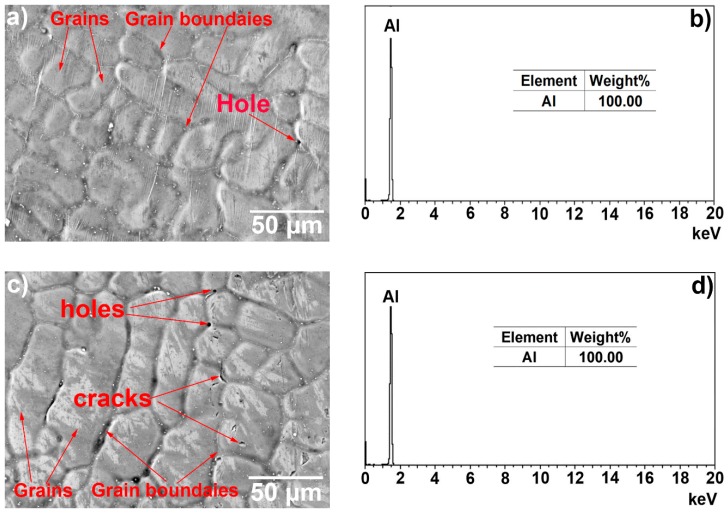
Surface morphologies and energy X-ray dispersive spectrometry (EDS) spectra and elemental compositions of hot-dip aluminised coatings on TA2 substrates: (**a**,**b**) T-1 and (**c**,**d**) T-2 samples, respectively.

**Figure 2 materials-12-00799-f002:**
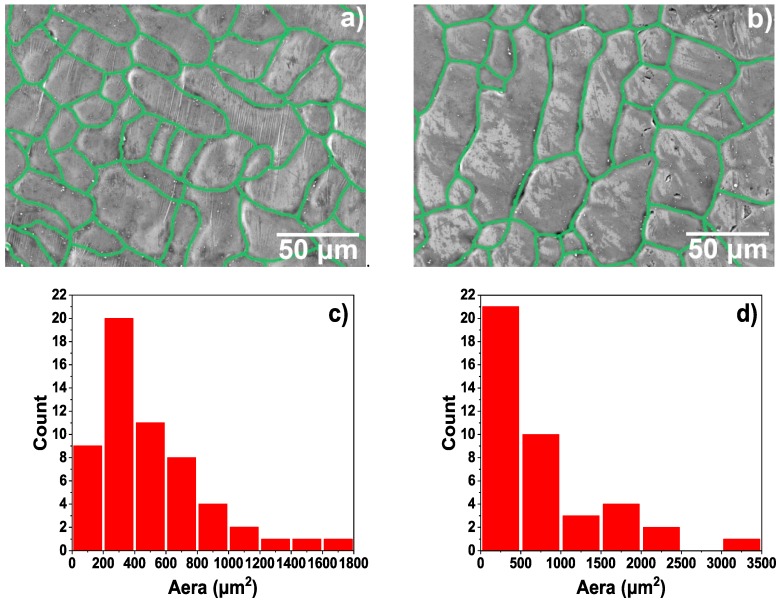
Grain boundary diagram and the grain distribution of hot-dip aluminised coatings obtained by image analysis: (**a**,**c**) T-1 sample, (**b**,**d**) T-2 sample.

**Figure 3 materials-12-00799-f003:**
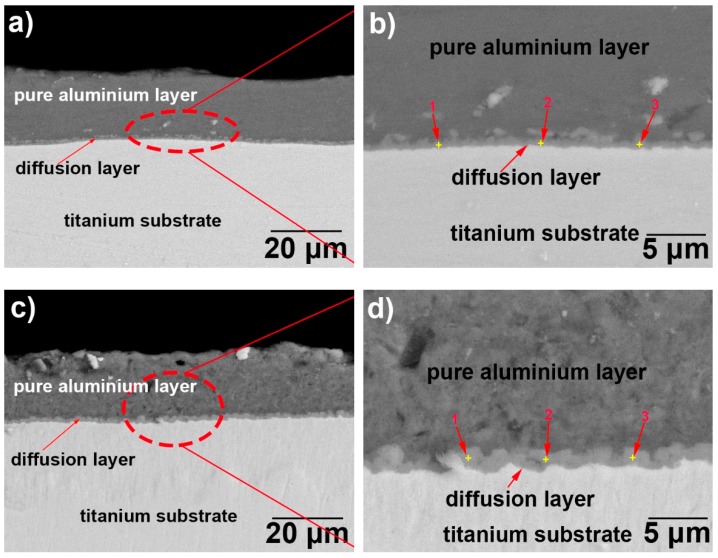
Cross-section of BSE morphologies of hot-dip aluminised coatings on TA2 substrate: (**a**,**b**) T-1 and (**c**,**d**) T-2 samples.

**Figure 4 materials-12-00799-f004:**
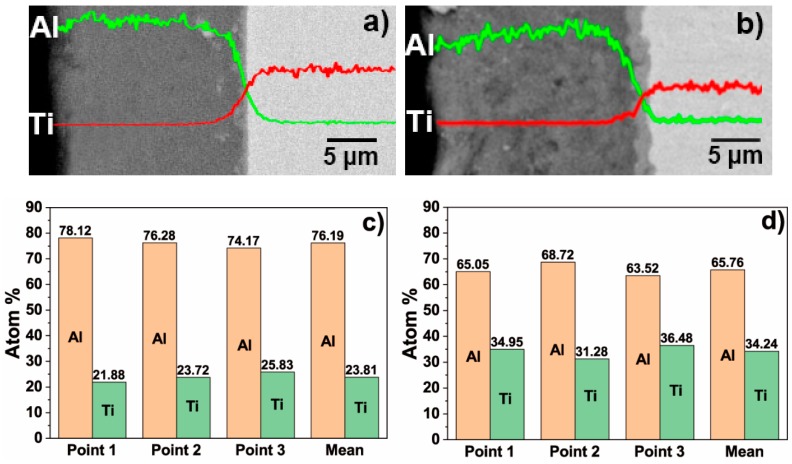
EDS element mapping of aluminium and titanium of (**a**) the T-1 and (**b**) T-2 samples; (**c**,**d**) the EDS point-scan analysis of hot-dip aluminised coatings formed on TA2 substrates for corresponding points in Figure 3b,d).

**Figure 5 materials-12-00799-f005:**
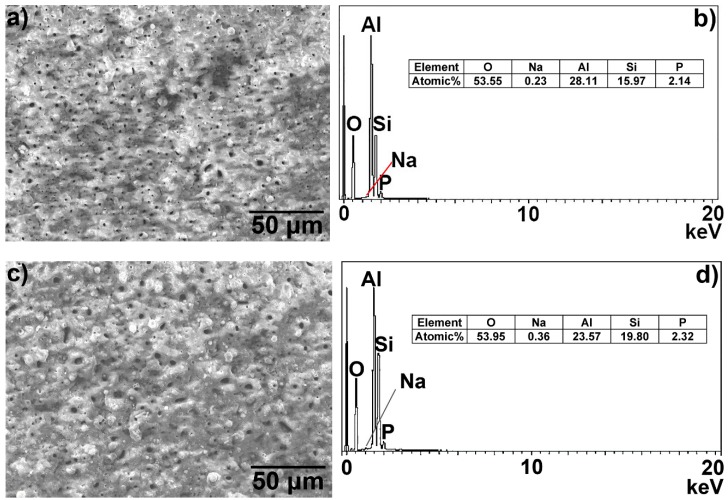
Surface morphology and composition of the MAO ceramic layer of the hot-dip aluminised coating on the TA2 substrate: (**a**,**b**) T-1 and (**c**,**d**) T-2 samples, respectively.

**Figure 6 materials-12-00799-f006:**
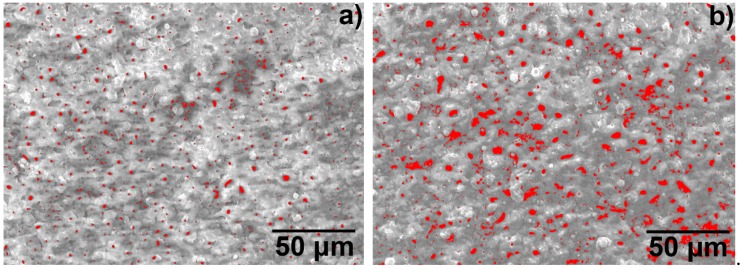
Image analysis results of the distribution and sizes of the holes in the MAO ceramic layer of the hot-dip aluminised coating on the TA2 substrate: (**a**,**c**) T-1 sample, (**b**,**d**) T-2 sample.

**Figure 7 materials-12-00799-f007:**
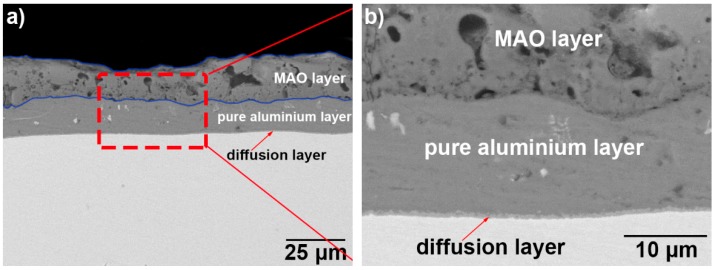
Cross-section morphology of the MAO ceramic layer of hot-dip aluminised coating on the TA2 substrate: (**a**,**b**) T-1 and (**c**,**d**) T-2 samples.

**Figure 8 materials-12-00799-f008:**
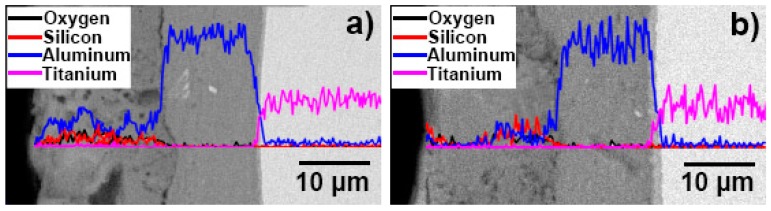
EDS line analysis of aluminium and titanium of the cross-section of the MAO layer formed on TA2 substrate after hot-dip aluminising: (**a**) T-1 sample, (**b**) T-2 sample.

**Figure 9 materials-12-00799-f009:**
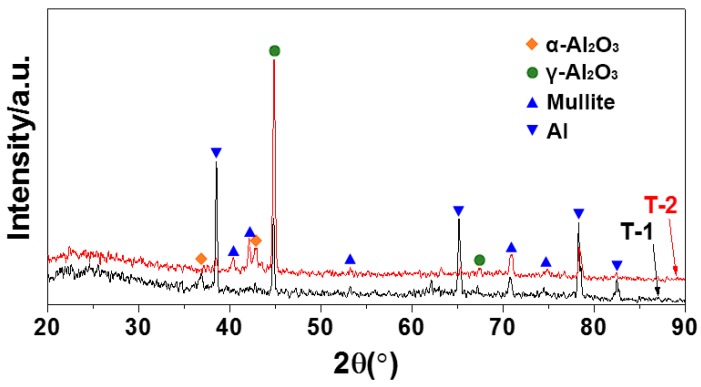
XRD spectra of MAO layers formed on TA2 substrates after hot-dip aluminizing.

**Figure 10 materials-12-00799-f010:**
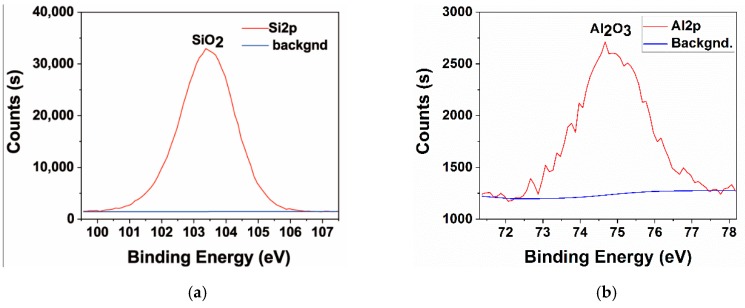
XPS spectrum of (**a**) the Si and (**b**) Al of T-1 sample.

**Figure 11 materials-12-00799-f011:**
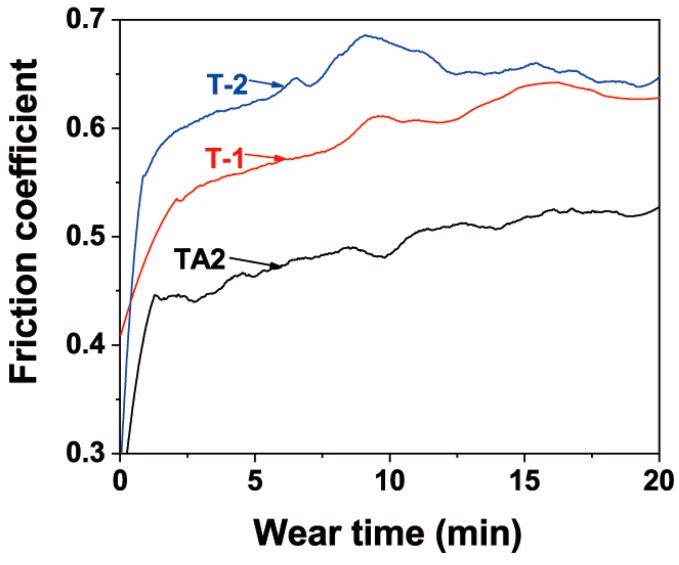
Friction coefficient curves of TA2, and T-1 and T-2 samples after MAO.

**Figure 12 materials-12-00799-f012:**
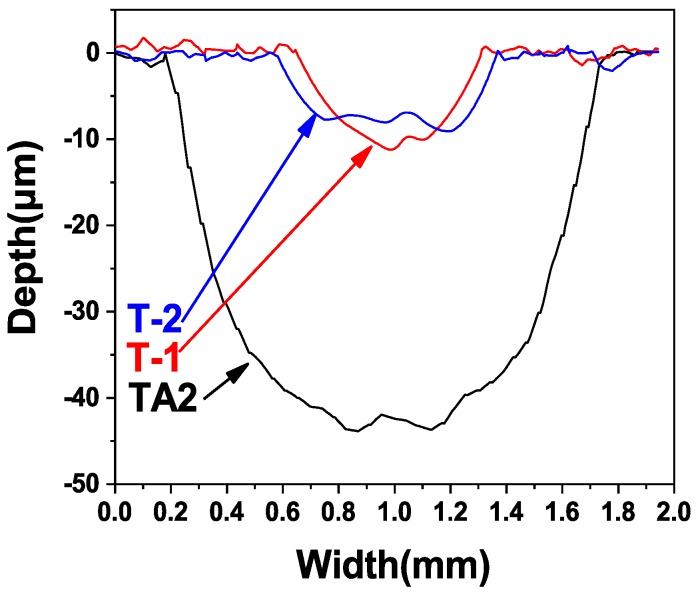
Cross-section wear scratch diagram of TA2, T-1 and T-2.

**Figure 13 materials-12-00799-f013:**
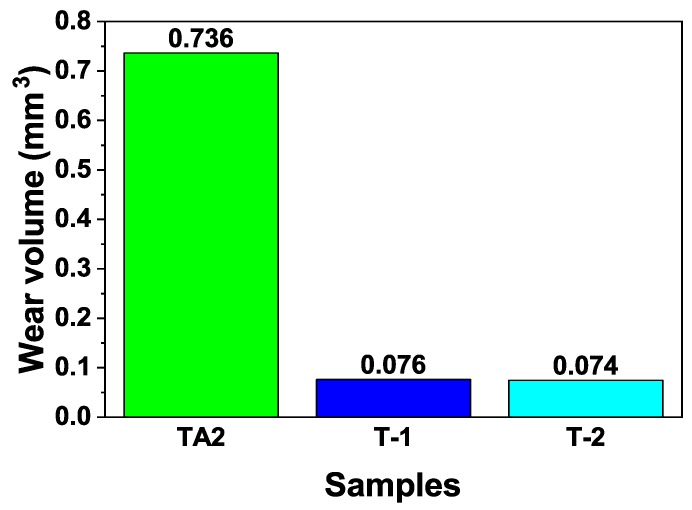
Wear volumes of TA2, T-1 and T-2.

**Figure 14 materials-12-00799-f014:**
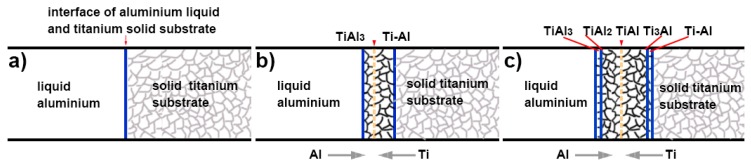
Schematic diagram of aluminium and titanium diffusion during the hot-dip aluminising process of the TA2 substrate. (**a**) wetting stage; (**b**) the initial stage of diffusion layer formation; (**c**) final stage of diffusion layer formation.

**Table 1 materials-12-00799-t001:** Process parameters of hot-dip aluminising and micro-arc oxidation (MAO).

Sample No.	Hot-Dip Aluminising			MAO			
	Temperature (°C)	Time (min)	Lifting Rate (cm·s^−1^)	Duty Cycle (%)	Pulse Frequency (Hz)	Voltage (V)	Time (min)
T-1	710	3	5	20	100	400–450	30
T-2	750	3	5	20	100	400–450	30

**Table 2 materials-12-00799-t002:** Measured wear scar width and depth and calculated wear volume.

Sample	Wear Scar Depth (mm)	Wear Scar Width (mm)	Wear Volume (mm^3^)
TA2	0.044	1.597	0.736
T-1	0.011	0.662	0.076
T-2	0.009	0.787	0.074

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
