# Peer review of "Morphology and Wear Resistance of Composite Coatings Formed on a TA2 Substrate Using Hot-Dip Aluminising and Micro-Arc Oxidation Technologies"

_materials, 2019, doi:10.3390/ma12050799_

Reviewer 1 Report

The paper is suitable to be published in the Journal, with some minor changes:

a) in the row 162 the sentence is not consistent

b) the part between 242 and 275 rows could be be summarised, with well pointed out arguments

c) the rows 328-330 could be missing, since all there data are presented in the table no 2

d) please correct minor errors (row 61 an, instead of a, in the row 69 the verb is missing, etc)

e) the concusions are too extended for a scientific journal, please try to make them more compact 

Author Response

Response to Reviewer 1 Comments

Point 1: in the row 162 the sentence is not consistent.

Response 1: in the row 162 the sentence was modified to "The surface topography of the T-2 sample was slightly rougher than that of the T-1 sample. "

Point 2: the part between 242 and 275 rows could be summarised, with well pointed out arguments.

Response 2: the rows 242-275 have been modified and the revised content is: 

Figure 5 illustrates the cross-sectional morphologies of the hot-dip aluminised samples after MAO treatment. The section of the ceramic layer with intermediate thickness was selected (Figure 5a and 5c). It can be seen that the obtained coatings exhibited a three-layered structure: the top layer was the ceramic layer, the second layer was the unoxidized pure aluminum layer, and the innermost layer was the diffusion layer obtained by hot-dip aluminizing. A part of the pure aluminum layer was oxidized to form a ceramic layer, while the rest of the pure aluminum layer and the diffusion layer kept unoxidized. The surfaces of the ceramic layers were rough and exhibited a certain degree of undulation. The average thickness was calculated using the measured maximum, minimum, and median thickness values. As shown in Figure 5a, the average thickness of the ceramic layer and the unoxidized pure aluminum layer with interdiffusion zone was respectively calculated to be 16.93 μm and 14.08 μm. In Figure 5c, the average thickness of the ceramic layer was calculated to be 20.99 μm using the same method, and that of the unoxidized pure aluminum together with the interdiffusion layer was 13.87 μm. Compared with Figure 2a and 2c, the MAO layer in Figure 5a and 5c was approximately four times thicker than that of the consumed pure aluminum. Moreover, it can be seen from Figures 5b and 5d that the MAO layers were denser than the pure aluminum one. The interface between the ceramic and pure aluminum layers was clear but undulant. This implied that the growth rate of the ceramic layer was different at the ceramic/aluminum interface. Additionally, a number of holes were observed in the cross-section of the ceramic layers, and this was determined by the nature of the MAO process. However, the difference between the substrate material and the MAO process will affect the distribution and size of these holes.

Point 3: the rows 328-330 could be missing, since all there data are presented in the table no 2.

Response 3: the rows 328-330 "The wear scar depths of the TA2, T-1, and T-2 samples were 0.044, 0.011, and 0.009 mm, respectively, the wear scar widths were 1.579, 0.662, and 0.787 mm, respectively, and the calculated wear volumes are 0.736, 0.076, and 0.074 mm3, respectively. " have been deleted.

Point 4: please correct minor errors (row 61 an, instead of a, in the row 69 the verb is missing, etc).

Response 4: in the row 61, "a" has been replaced with "an". In the row 69, the verb has been added.

Point 5: the concusions are too extended for a scientific journal, please try to make them more compact.

Response 5: the concusions have been streamlined and the conclusions after streamlining are as follows:

1) An aluminium coating was deposited on the titanium substrate using the hot-dip aluminising technique, and it presented a two-layered structure with a pure aluminium layer outside and a diffusion Ti-Al layer inside. The Ti-Al diffusion layer was well bonded to the pure aluminium layer and pure titanium substrate.

2) A ceramic layer mainly consisted of α-Al2O3, γ-Al2O3, mullite, and unoxidised aluminium was obtained after the treatment of the hot-dip aluminised layer. The thicknesses of the aluminium layers consumed during MAO were approximately one quarter of the thicknesses of the obtained ceramic layers. Moreover, the content of the ceramic phase in the ceramic layer obtained after the 750 °C hot-dip aluminised sample underwent MAO was higher than that of the 710 °C hot-dip aluminised sample. The interface between the ceramic and residual aluminium layers was well combined.

3) The wear resistance of the samples was significantly improved after ceramic layers forming on the surfaces of pure titanium substrates. The friction coefficient of the pure titanium sample was determined to be 0.52, while the friction coefficients of the samples subjected to MAO after hot-dip aluminising at 710 and 750 °C were 0.62 and 0.65, respectively. The corresponding calculated wear volumes of the three samples were respectively calculated to be 0.736, 0.076, and 0.074 mm3.

Reviewer 2 Report

The manuscript deals with hot-dip aluminisig of a titanium substrate and MAO modification of the aluminised surface. The manuscript is nicely written, the introductory part is good, all descriptions are given on sufficient level of details, technically on high level. However, the manuscript resembles a technical report on what was done and observed rather than a scientific article although the amount of the experimental results should allow better discussion and formulation of results.

Evaluation of micrographic properties (morphology?) is missed in the text – the images of samples observed from the top are very weakly described (if at all) and definitely not quantified. Does it play any role?  

Captions of all SEM figures have to indicate the mode of detection (backscattered or secondary electrons??)

Lines 269 and 270 “In addition, it can be seen from Figures 5b) and d) that the MAO layers were more  dense than the pure aluminium one,” … I cannot see it. What density?

Lines 272 - 275 there are no data on evaluation of the distribution and size off the holes in MAO layers. Support the statement somehow. Image analysis? What are the certain numbers?

Line 277 Figure 6. SEM image is combined with EDS analysis. It can be seen that the image has much sharper borders between phases than the curves for elemental composition. (therefore the info about the detection mode would be usefull). Why is this? Is this due to the spatial resolution of the EDS (as the reviewer guesses) or is the composition of such diffuse character (?)  The first case would support your ideas about intermetallic compositons (stoichiometries) in discussion.

Lines 336-394. These discussion is nice, however, there is only one reference [41]  and the whole text is just speculating or repeating what was written there, although the formulation of this text is much better understandable than the original one, and Figure 12 is very instructive. This will be a verz nice part of a textbook, but there is actually no discussion of obtained results (!) in this large paragraph. One reference is in Line 353 and following, however it refers to the Figure 12 which does not contain any original data. … It is a nice description of the idea depicted in Figure 12. So far, there is only one real reference to the presented results i.e. Lines 387-389… “It can be observed from Figure 3 that for the same hot-dip time, as the temperature increased, the thickness of the diffusion layer significantly increased.” … The discussion should be based substantially on presented results. This part of the manuscript must be rewritten, actually, the results needs to be re-discussed.

Lines 407-432. MMAO is discussed, or is not discussed? Please, make clear what you are aiming on. It is confusing. Are the MAO processes the same or are they different?   Rewriting is necessary.

Moreover, a discussion of correlation between wear test performance and morphology (micrographic properties) is missed although the reviewer would guess this is the main aim of the article. Shouldn’t be this the core of the manuscript?  

Lines 433 and following. Rather than Conclusions a Summary of what was done and measured is presented. This chapter should be possibly renamed (such as Summary and conclusions or so….). By the way, aren’t there any observable trends in obtained results, any general lesson drawn from the work performed?  Point 3 (lines 444-447) is too obvious and should be removed.  The last sentence in the conclusions (“Therefore the wear resistance of the samples significantly improved after ceramic layers were formed on the surfaces of pure titanium substrates.”) is in some mismatch to end of the discussion section lines 427-432 (“In addition, the wear resistance values of the ceramic layers formed by MAO were significantly higher than that of the pure titanium sample, mainly owing to the presence of ceramic phases such as Al2O3 and mullite in the ceramic layers. The mechanism for improving the wear resistance was basically the same as that of the aluminium substrate surface: MAO was used to form the oxide ceramic coating, and will not be discussed here.”) How is it? Unclear discussion resulted into unclear conclusions.

What is totally missed in chapters Results, Discussion and Conclusion is a comparison with results obtained by other methods in references, otherwise no reader will know about the originality and novelty in presented results, and will not be able to judge about its real importance in comparison with other methods or results obtained by other authors.

Notes of lesser importance:

Line 2. Micrographic properties? Maybe „Morphology and wear resistence….“ sounds better.

Line 18. “The phase structure of the MAO layers was studied using X-ray diffraction and X-ray photoelectron spectroscopy.” I think, the phase structure was studied by X-ray diffraction. XPS was used for study of composition – which is not the same as phase structure. This sentence should be changed.

Lines 175, 178, and on many other places in the text .. every average number is presented with surprizing precision e.g. the thickness 0.61 um (line 178). According to my opinion, it is evident in Figure 2, that the diffusion layer varies in thickness much largely than in tens of nm. Similarly, all other thicknesses! The estimations should be rounded with respect to the variation of the values.

Line 202 and many others. Similar problem with compositions. However, it can be, that only one measurement was recorded, then nobody has information about the variability in composition. If there is more than one measurement available for one sample, please, indicate the variability in composition by proper rounding of the numbers.

Line 235 “phosphorus” instead of “phosphorous”

Author Response

Response to Reviewer 2 Comments

Point 1: the manuscript deals with hot-dip aluminisig of a titanium substrate and MAO modification of the aluminised surface. The manuscript is nicely written, the introductory part is good, all descriptions are given on sufficient level of details, technically on high level. However, the manuscript resembles a technical report on what was done and observed rather than a scientific article although the amount of the experimental results should allow better discussion and formulation of results.

Response 1: In the article, the discussion and analysis of the test results are added, and more test results are cited in the analysis discussion.

Point 2: evaluation of micrographic properties (morphology?) is missed in the text - the images of samples observed from the top are very weakly described (if at all) and definitely not quantified. Does it play any role?

Response 2: The description and analysis of the surface morphology of the hot-dip aluminising layer and the micro-arc oxidation layer were added, and the surface morphology was analyzed by image analysis(lines 186-208 and 264-279). The image analysis result of the surface morphology of the hot-dip aluminized layer and the image analysis of the micro-arc oxidation surface topography are added(figures 2 and 5), the original figures 2, 3, 4, 5, 6, 7, 8, 9, 10, 11, 12 are correspondingly changed to Figures 3, 4, 5, 7, 8, 9, 10, 11, 12, 13, 14).

Point 3: captions of all SEM figures have to indicate the mode of detection (backscattered or secondary electrons??)

Response 3: captions of all SEM figures have been indicate the mode of detection.

Point 4: lines 269 and 270 “In addition, it can be seen from Figures 5b) and d) that the MAO layers were more dense than the pure aluminium one,” … I cannot see it. What density?

Response 4: lines 269 and 270 “In addition, it can be seen from Figures 5b) and d) that the MAO layers were more dense than the pure aluminium one,” have been revised as “Moreover, it can be seen from Figures 7b and 7d that MAO layer were well bonded to the pure aluminum layerand no obvious microscopic defects such as holes and cracks were observed at the interface.”297-299 lines

Point 5: lines 272 - 275 there are no data on evaluation of the distribution and size off the holes in MAO layers. Support the statement somehow. Image analysis? What are the certain numbers?

Response 5: The size and distribution of the pores in the MAO layer were analyzed by image analysis method, and the distribution map and distribution statistics of the pores were increased (figures 5, lines 264-279).

Point 6: line 277 Figure 6. SEM image is combined with EDS analysis. It can be seen that the image has much sharper borders between phases than the curves for elemental composition. (therefore the info about the detection mode would be usefull). Why is this? Is this due to the spatial resolution of the EDS (as the reviewer guesses) or is the composition of such diffuse character (?) The first case would support your ideas about intermetallic compositons (stoichiometries) in discussion.

Response 6: Because Figure 6 (Figure 8 in the revised manuscript) is a backscattered electronic mode images, it is mainly caused by the resolution of the EDS.

Point 7: lines 336-394. These discussion is nice, however, there is only one reference [41] and the whole text is just speculating or repeating what was written there, although the formulation of this text is much better understandable than the original one, and Figure 12 is very instructive. This will be a verz nice part of a textbook, but there is actually no discussion of obtained results (!) in this large paragraph. One reference is in Line 353 and following, however it refers to the Figure 12 which does not contain any original data. … It is a nice description of the idea depicted in Figure 12. So far, there is only one real reference to the presented results i.e. Lines 387-389… “It can be observed from Figure 3 that for the same hot-dip time, as the temperature increased, the thickness of the diffusion layer significantly increased.” … The discussion should be based substantially on presented results. This part of the manuscript must be rewritten, actually, the results needs to be re-discussed.

Response 7: Lines 336-394. The conclusion has been rewritten, the reference has been added, and the analysis of the test results has been strengthened. Modified to lines 367-444.

Point 8: lines 407-432. MAO is discussed, or is not discussed? Please, make clear what you are aiming on. It is confusing. Are the MAO processes the same or are they different? Rewriting is necessary.

Response 8: Lines 407-432, has been rewritten, increase the analysis of the characteristics of the MAO layer (lines 457-497).

Point 9: moreover, a discussion of correlation between wear test performance and morphology (micrographic properties) is missed although the reviewer would guess this is the main aim of the article. Shouldn’t be this the core of the manuscript?

Response 9: Added paragraphs to discuss the relationship between microtopography and wear testing (lines 480-497).

Point 10: lines 433 and following. Rather than Conclusions a Summary of what was done and measured is presented. This chapter should be possibly renamed (such as Summary and conclusions or so….). By the way, aren’t there any observable trends in obtained results, any general lesson drawn from the work performed? Point 3 (lines 444-447) is too obvious and should be removed. The last sentence in the conclusions (“Therefore the wear resistance of the samples significantly improved after ceramic layers were formed on the surfaces of pure titanium substrates.”) is in some mismatch to end of the discussion section lines 427-432 (“In addition, the wear resistance values of the ceramic layers formed by MAO were significantly higher than that of the pure titanium sample, mainly owing to the presence of ceramic phases such as Al2O3 and mullite in the ceramic layers. The mechanism for improving the wear resistance was basically the same as that of the aluminium substrate surface: MAO was used to form the oxide ceramic coating, and will not be discussed here.”) How is it? Unclear discussion resulted into unclear conclusions.

Response 10: The 433 lines and the following, the conclusions have been rewritten. The conclusions after rewriting are as follows:

1) An aluminium coating was deposited on the titanium substrate using the hot-dip aluminising technique, and it presented a two-layered structure with a pure aluminium layer outside and a diffusion Ti-Al layer inside. The Ti-Al diffusion layer was well bonded to the pure aluminium layer and pure titanium substrate.

2) A ceramic layer mainly consisted of α-Al2O3, γ-Al2O3, mullite, and unoxidised aluminium was obtained after the treatment of the hot-dip aluminised layer. The thicknesses of the aluminium layers consumed during MAO were approximately one quarter of the thicknesses of the obtained ceramic layers. Moreover, the content of the ceramic phase in the ceramic layer obtained after the 750 °C hot-dip aluminised sample underwent MAO was higher than that of the 710 °C hot-dip aluminised sample. The interface between the ceramic and residual aluminium layers was well combined.

3) The wear resistance of the samples was significantly improved after ceramic layers forming on the surfaces of pure titanium substrates. The friction coefficient of the pure titanium sample was determined to be 0.52, while the friction coefficients of the samples subjected to MAO after hot-dip aluminising at 710 and 750 °C were 0.62 and 0.65, respectively. The corresponding calculated wear volumes of the three samples were respectively calculated to be 0.736, 0.076, and 0.074 mm3.

Point 11: what is totally missed in chapters Results, Discussion and Conclusion is a comparison with results obtained by other methods in references, otherwise no reader will know about the originality and novelty in presented results, and will not be able to judge about its real importance in comparison with other methods or results obtained by other authors.

Response 11: In the discussion section, a comparison with other methods has been added, as follows:

A ceramic coating containing aluminum and silicon oxides can be prepared on the surface of a pure titanium substrate by a combination of hot dip aluminising and micro-arc oxidation processes using a silicate system electrolyte. The micro-arc oxidation process directly micro-arc oxidation on the titanium and its alloys can only prepare ceramic coatings of titanium oxide and silicon oxide, and the coating growth efficiency and coating compactness are lower than that of aluminum micro-arc oxidation. In addition, because the hardness of titanium oxide is lower than that of aluminum oxide, the coating wear resistance is not as good as that of the coating mainly composed of aluminium oxide.

The pure aluminum layer is first deposited on the surface of titanium by magnetron sputtering, and then the deposited aluminum layer is oxidized to form a ceramic coating by micro-arc oxidation technology. It can also prepare composition coating composed of a pure aluminum layer and a ceramic layer on the titanium surface. However, compared with the process adopted herein, the deposition coating has no diffusion layer, the coating does not form a metallurgical bond, and the magnetron sputtering deposition needs to be completed in a vacuum chamber, and the preparation process is relatively complicated, and The deposition efficiency of the coating is also lower.

Point 12: line 2. Micrographic properties? Maybe “Morphology and wear resistence….” sounds better.

Response 12: line 2 "Micrographic properties and wear resistance…." has been changed as "Morphology and wear resistence…. ".

Point 13: line 18. “The phase structure of the MAO layers was studied using X-ray diffraction and X-ray photoelectron spectroscopy.” I think, the phase structure was studied by X-ray diffraction. XPS was used for study of composition – which is not the same as phase structure. This sentence should be changed.

Response 13: in the line 18, "The phase structure of the MAO layers was studied using X-ray diffraction and X-ray photoelectron spectroscopy." have been revised as " The phase structure of the MAO layers was studied using X-ray diffraction. The surface composition of MAO layer was studied by X-ray photoelectron spectroscopy."

Point 14: lines 175, 178, and on many other places in the text .. every average number is presented with surprizing precision e.g. the thickness 0.61 um (line 178). According to my opinion, it is evident in Figure 2, that the diffusion layer varies in thickness much largely than in tens of nm. Similarly, all other thicknesses! The estimations should be rounded with respect to the variation of the values.

Response 14: lines 175, 178, and on many other places in the text All values regarding thickness have been rounded off, lines 196, 199, 204, 205, 294, 295, 296).

Point 15: line 202 and many others. Similar problem with compositions. However, it can be, that only one measurement was recorded, then nobody has information about the variability in composition. If there is more than one measurement available for one sample, please, indicate the variability in composition by proper rounding of the numbers.

Response 15: The EDS point scan test of the diffusion layer component of the hot-dip aluminising layer was carried out. In the middle of the diffusion layer, the components at three different positions were measured, the average value was calculated, and the data comparison chart was drawn and analyzed.

Point 16: line 235 “phosphorus” instead of “phosphorous”

Response 16: in the line 235, " phosphorous " has been replaced with " phosphorus.

Reviewer 3 Report

This work is focused on to use hot-dip aluminising and MAO to indirectly 99 generate Al2O3 ceramic coatings on the surface of titanium and titanium alloys, thus, improving their 100 wear resistance. It shows some interesting results and discussions. It can be considered for publication after addressing the following comments:

1) Please correct " ….surface comprehensive performance tester(MFT-4000, Lanzhou Institute of Chemical Physics, 145 Lanzhou, China),…." as " surface comprehensive performance tester (MFT-4000, Lanzhou Institute of Chemical Physics, 145 Lanzhou, China) ".

2) Please correct " …. where b) is a partial enlargement of a)) was dense and continuous, and presented no defects, such 173 as cracks or micropores." as " where b) is a partial enlargement of a) was dense and continuous, and presented no defects, such 173 as cracks or micropores.".

3) Please correct " …. is a partial enlargement of c)) was relatively smaller than that of the hot-dip aluminium 181 layered obtained at 710 °C. " as " is a partial enlargement of c) was relatively smaller than that of the hot-dip aluminium 181 layered obtained at 710 °C.".

4) Please correct " …. form a solid solution with titanium, as illustrated in Figure 12 b)." as " form a solid solution with titanium, as illustrated in Figure 12 b. ".

5) Please correct " …. Ti3Al, and other titanium-aluminium intermetallic compounds were located at the centre of the 378 diffusion layer, as shown in Figure 12c)." as " Ti3Al, and other titanium-aluminium intermetallic compounds were located at the centre of the 378 diffusion layer, as shown in Figure 12c.".

Author Response

Response to Reviewer 2 Comments

Point 1: Please correct " …. surface comprehensive performance tester(MFT-4000, Lanzhou Institute of Chemical Physics, 145 Lanzhou, China),…." as " surface comprehensive performance tester (MFT-4000, Lanzhou Institute of Chemical Physics, 145 Lanzhou, China) ".

Response 1: in the row 145 sentence " …. surface comprehensive performance tester(MFT-4000, Lanzhou Institute of Chemical Physics, Lanzhou, China), " has been modified to "…. surface comprehensive performance tester (MFT-4000, Lanzhou Institute of Chemical Physics, Lanzhou, China). "

Point 2: Please correct " …. where b) is a partial enlargement of a)) was dense and continuous, and presented no defects, such 173 as cracks or micropores." as " where b) is a partial enlargement of a) was dense and continuous, and presented no defects, such 173 as cracks or micropores.".

Response 2: the sentence in the 173 row ".... where b) is a partial enlargement of a)) was dense and continuous, and presented no defects, such as cracks or micropores." was modified by the author as " The aluminium layer obtained after 3 min of Hot-dip aluminising at 710 °C (sample T-1, Figures 2a and b, where figure 2b is a partial enlargement of figure 2a) was dense and continuous, and presented no defects, such as cracks or micropores.

Point 3: Please correct " …. is a partial enlargement of c)) was relatively smaller than that of the hot-dip aluminium 181 layered obtained at 710 °C. " as " is a partial enlargement of c) was relatively smaller than that of the hot-dip aluminium 181 layered obtained at 710 °C.".

Response 3: the sentence in the 173 row " …. is a partial enlargement of c)) was relatively smaller than that of the hot-dip aluminium 181 layered obtained at 710 °C. " was modified as " The density of the pure aluminium layer obtained after 3 min of hot-dip aluminising at 750 °C (sample T-2, Figures 2c and d, where figure 2d is a partial enlargement of figure 2c) was relatively smaller than that of the hot-dip aluminium layered obtained at 710 °C. "

Point 4: Please correct " …. form a solid solution with titanium, as illustrated in Figure 12 b)." as " form a solid solution with titanium, as illustrated in Figure 12 b.".

Response 4: the sentence" …. form a solid solution with titanium, as illustrated in Figure 12 b). " modified to " …. form a solid solution with titanium, as illustrated in Figure 12b."

Point 5: Please correct " …. Ti3Al, and other titanium-aluminium intermetallic compounds were located at the centre of the 378 diffusion layer, as shown in Figure 12c)." as " Ti3Al, and other titanium-aluminium intermetallic compounds were located at the centre of the 378 diffusion layer, as shown in Figure 12c.".

Response 5: the sentence in the row 378" …. Ti3Al, and other titanium-aluminium intermetallic compounds were located at the centre of the diffusion layer, as shown in Figure 12c)." was modified as " Ti3Al, and other titanium-aluminium intermetallic compounds were located at the centre of the diffusion layer, as shown in Figure 12c.".

The author removed the parentheses from all the drawing numbers in the article, such as figure 2 b) modified to figure 2b.

Round  2

Reviewer 2 Report

Apologize for the delay with the second review, I checked the response of the authors to my first review and they addressed all my comments in a very good way, I am sure, I would have no other comments to the manuscript and would recommend to publish it. Hope at least, my first
review helped the authors to improve the quality of the manuscript.